# The Antimicrobial and Antibiofilm Potential of New Water-Soluble Tris-Quaternary Ammonium Compounds

**DOI:** 10.3390/ijms241310512

**Published:** 2023-06-22

**Authors:** Nikita A. Frolov, Mary A. Seferyan, Anvar B. Valeev, Evgeniya A. Saverina, Elena V. Detusheva, Anatoly N. Vereshchagin

**Affiliations:** 1N. D. Zelinsky Institute of Organic Chemistry, Russian Academy of Sciences, Leninsky Prospect 47, 119991 Moscow, Russia; nfrolov@ioc.ac.ru (N.A.F.); marysev@ioc.ac.ru (M.A.S.); valeev@ioc.ac.ru (A.B.V.); esaverina94@gmail.com (E.A.S.); detushevaev@obolensk.org (E.V.D.); 2Higher Chemical College of the Russian Academy of Sciences, D. Mendeleev University of Chemical Technology of Russia, Miusskaya Square 9, 125047 Moscow, Russia; 3Laboratory of Biologically Active Compounds and Biocomposites, Tula State University, Lenin Prospect. 92, 300012 Tula, Russia; 4State Research Center for Applied Microbiology and Biotechnology, 142279 Obolensk, Russia

**Keywords:** quaternary ammonium compounds, pyridinium salts, disinfectants, antiseptics, antibacterial activity, antifungal activity, antibiofilm activity, water soluble, cyanuric acid derivatives, bacterial resistance

## Abstract

The invention and innovation of highly effective antimicrobials are always crucial tasks for medical and organic chemistry, especially at the current time, when there is a serious threat of shortages of effective antimicrobials following the pandemic. In the study presented in this article, we established a new approach to synthesizing three novel series of bioactive water-soluble tris-quaternary ammonium compounds using an optimized one-pot method, and we assessed their antimicrobial and antibiofilm potential. Five pathogenic microorganisms of the ESKAPE group, including highly resistant clinical isolates, were used as the test samples. Moreover, we highlighted the dependence of antibacterial activity from the hydrophilic–hydrophobic balance of the QACs and noted the significant performance of the desired products on biofilms with MBEC as low as 16 mg/L against bacteria and 8 mg/L against fungi. Particularly notable was the high activity against *Pseudomonas aeruginosa* and *Acinetobacter baumannii*, which are among the most resilient bacteria known. The presented work will provide useful insights for future research on the topic.

## 1. Introduction

The development of new highly effective antimicrobials that combine a wide range of activity, environmental safety, and relatively low toxicity is extremely important for medical and organic chemistry [1], especially now, when the world is struggling with the pandemic aftermath. Even though the number of registered COVID-19 death cases is at the lowest point since 2020 [2], the long-term consequences cannot be ignored.

One of the major post-pandemic concerns is the development of antimicrobial resistance (AMR) to commercial biocides due to the increased frequency and irresponsibility regarding self-medication [3,4,5]. This led to a sharp decrease in the effectiveness of therapeutic and preventive measures in hospitals and contributed to the spread of lethal infections. The described problem was extremely severe and relevant, even before the pandemic [6,7]. Moreover, according to the 2022 CDC (Centers for Disease Control and Prevention) report, the situation regarding AMR-related infections has worsened by at least 15% since 2019 [8]. Thus, health care facilities may face a serious shortage of effective antimicrobials in the coming years. Therefore, there is a vital need for the discovery of new antibacterial, antifungal, and antibiofilm agents and for the innovation of the existing agents.

Quaternary ammonium compounds (QACs) are a class of versatile and effective antimicrobial agents which have been widely used in various fields, including healthcare, food processing, and agriculture [9]. They are a type of cationic surfactant that possesses disinfectant, sanitizing, and deodorizing properties and which can be applied as agents for the prevention and eradication of pathogenic biofilms [10,11]. QACs consist of one or more positively charged nitrogen atoms with four substituents and are divided into four main groups, namely mono-, bis-, multi-, and poly-cationic compounds [12,13,14]. Some of the most recognizable disinfectants and antiseptics among commercial QACs are benzalkonium chloride [15], miramistin [16], didecyldimethylammonium chloride [17], cetylpyridinium chloride [18], and octenidine dichloride [19] (Figure 1).

While mono- and bis-QACs are rather well-researched, research into biologically active multi-QACs is drastically underdeveloped. The tris-QACs (especially with a pyridine core and three alkyl tails) as biocides have been studied very little. Currently, there are only a few works covering this subject [20,21,22,23,24] (Figure 2).

The Wuest and Minbiole group devoted multiple publications to the topic of QACs, including tris-QACs, as antibacterials. They highlighted the significant advantage of tris-QACs with pyridinium core over mono-QACs (Figure 2B, left) [23]. The developed compounds outperformed many commercial disinfectants in activity and were among the most effective biocides in the QAC library [21]. Other reported examples of the tris- and multi-QAC antibacterial activity revealed that an increased number of charged nitrogen cores and alkyl tails are favorable mainly for their biocidal effects [20,21,22,25,26].

With these hypotheses in mind, we conducted a study devoted to a new type of tris-QACs based on an alkylcyanuric spacer [24] (Figure 2B, right). The study showed promising results, with modest activity against several pathogenic bacterial strains. However, the desired compounds possessed poor water solubility, which is an important factor for modern antiseptics. Water-soluble compounds can be used in water-based biocide compositions, which are significantly less harmful than alcohol compounds [27].

Herein, we report a synthesis and microbiological evaluation of novel water-soluble pyridinium tris-QACs on a cyanuric scaffold with ether fragments (Figure 3, Appendix A). The synthesized compounds were tested on a wide range of pathogenic microorganisms of the ESKAPE group [28,29,30], including biofilms and highly resistant clinical isolates. The established approach provided a new generation of effective antimicrobial agents from available resources.

## 2. Results and Discussion

### 2.1. Synthesis of Novel Tris-QACs

In preparing a wide range of new tris-QACs with incorporated ether groups, we sought to exploit a previously developed synthetic approach [24]. Firstly, spacer parts were constructed from available glycols within the two-stage method (Figure 1A). Synthesis of the cyanuric spacer structure (**3a**–**c**) began with the conversion of glycols into chlorides (**2a**–**c**) through a nucleophilic substitution.

Accordingly, exposure to an excess of thionyl chloride at reflux in either benzene or toluene with a pyridine additive, followed by extraction and distillation, led to almost quantitative yields of the corresponding chlorides **2a**–**c**.

Next, we performed the synthesis of the tris-chloroderivatives **3a**–**c**, following a known procedure with slight modifications [31]. The reference method featured a two-pot synthesis of cyanuric triamide and triester through trisodium salt formation with further alkylation in dimethylformamide (DMF). However, the described method was not suitable for our platforms. The present condition produced a complex mixture of by-products and 0 to 5% yields of the desired scaffolds (Figure 2A). To improve the yields, we modified the reaction procedure to a one-pot synthesis using sodium hydride and hexamethylphosphoramide (HMPA). When the reaction was performed under optimized conditions, the yield of **3a** increased 12-fold, and by-products were not detected (Figure 2B). The products were purified by flash chromatography.

Alkylaminopyridines (**6a**–**c**, core–tail fragment) were synthesized according to the known method used in our previous study (Figure 1B) [24]. Final salts were prepared via alkylation (quaternization), also known as the Menshutkin reaction [32]. The reaction proceeds in various polar solvents (such as acetone, acetonitrile, methyl isobutyl ketone, DMF, and DMSO) under heating. In the case of the novel tris-QACs, optimal conditions with the highest yields were observed using butanol-1 (Figure 1C).

Synthetic procedures and characterization data for all compounds are represented in the Section 3.

### 2.2. Antibacterial Activity of Novel Tris-QACs

Firstly, the prepared tris-QACs were screened for their antibacterial properties against the reference ESKAPE pathogens (Gram-positive *Staphylococcus. aureus* ATCC 43,300 (***Sa***), Gram-negative *Escherichia coli* ATCC 25,922 (***Ec***), *Klebsiella pneumoniae* ATCC 70,060 (***Kp***), *Pseudomonas aeruginosa* ATCC 27,853 (***Pa***), and *Acinetobacter baumannii* ATCC 15,308 (***Ab***)) in planktonic form. The bioactivities of the prepared cationic compounds were assessed via minimum inhibitory concentration (MIC, mg/L) and minimum bactericidal concentration (MBC, mg/L). The purpose of the primary screening was to define the most active series of novel tris-QACs for their further extended microbiological analysis. Thus, three compounds from each series (with C8, C10, and C12 alkyl tails) were tested (Table 1).

The results showed that Series **7** exhibited higher potency against a broad spectrum of bacteria than the rest of the tris-QACs, particularly compound **7c**, which demonstrated a promising range of activity, with 16–32 mg/L minimal bactericidal concentration on the Gram-negative strains. Series **8** displayed modest biocidal action, with 1 to 5 dilution differences compared with the best MBCs values, whereas Series **9** showed very poor performance. Therefore, Series **7** was selected for further microbiological evaluation.

The broad microbiological study comprised the investigation of wider tris-QACs (C6-C12 alkyl tails) and pathogen ranges, including five multidrug-resistant clinical isolates (E. coli B-3421/19, *K. pneumoniae* B-2523/18, S. aureus B-8648, *P. aeruginosa* B-2099/18, and *A. baumannii* B-2926/18), as well as the yeast-like fungi (*Candida albicans* ATCC 10,231 and *Candida albicans* AV-85), in planktonic and biofilm forms. The bioactivities against biofilms were assessed via the minimum biofilm inhibitory concentration (MBIC, mg/L) and minimum biofilm eradication concentration (MBEC, mg/L). The commercial mono-QACs benzalkonium chloride (BAC) and cetylpyridinium chloride (CPC), bis-QAC octenidine dichloride (OCT) (Figure 1), and a hit compound from a previous work TQAC-8 (Figure 2) [24] were tested as reference samples (Table 2).

The presented results provided impactful insights on the biocide properties of the novel tris-QACs. Activities of ether-linked salts 7a–g on the planktonic bacteria were more prominent than the commercial mono-QACs (BAC and CPC) and the previously synthesized series (with TQAC-8 as the lead compound). Therefore, the introduction of the ether components in the spacer segment greatly improved the antibacterial activity, with a 4-fold lower MBC on *S. aureus* and 2- to 32-fold lower MBCs on the Gram-negative bacteria. However, the biocidal effect of the novel tris-QACs was poorer in comparison with the lead commercial bis-QAC (OCT).

The most encouraging results were indicated on both the reference and clinical *A. baumanii* and *P. aeruginosa* strains, where inhibition and bactericidal activities of **7c**–**e** were the best among all the tested compounds, including the commercial compounds.

The overall outcomes of the antibacterial assay concerning the planktonic forms are consistent with the literature data and trends on the topic. Thus, MIC and MBC values are correlated with the alkyl tail range, with C8-C10 (**7c**–**e**) as the optimal diapason. The cut-off effect was also noted when examining the Gram-negative pathogens: *Klebsiella pneumoniae* ATCC 70,060 (16-fold, from C11 to C12), *K. pneumoniae* B-2523/18 (16-fold, from C11 to C12), *Pseudomonas aeruginosa* ATCC 27,853 (16-fold, from C10 to C12), and *P. aeruginosa* B-2099/18 (8-fold, from C10 to C11). The cut-off effect was described more than 100 years ago and was stated to be a sharp decrease in biological activity of the surfactants after reaching the critical point of the alkyl chain length in a series of homologous compounds [33].

Meanwhile, the antibiofilm assay results indicated the great potential of the new tris-QACs. The antibiofilm activity of compound **7c** outperformed more than 150 biocides that we synthesized and tested, making it one of our top compounds to date. Moreover, **7b** and **7c** demonstrated the ability to inhibit and remove *A. baumanii* and *P. aeruginosa* biofilms, which are among the most resilient bacteria known [34,35]. Noteworthy, the cut-off effect was evident in almost all the strains tested, except for two clinical isolates. Furthermore, the decrease in activity was observed in the compounds with shorter alkyl tails compared with the planktonic samples (C8-C10 for biofilms and C10-C12 for planktonic cells). This observation is supported by examples in the literature, where the best MBEC values were achieved with shorter alkyl lengths [20,36].

The investigation of the antifungal activity concluded our microbiological study. Here, the optimal fungicidal effect was achieved in a wide tris-QACs range (**7c**–**7f**, C8–C11). The best compounds (**7c** and **7f**) were slightly more active than the commercial antiseptic octenidine dichloride. Unexpectedly, the cut-off effect was the same on both the planktonic cells and the biofilms (Table 3), which is inconsistent with the abovementioned examination of bacterial pathogens.

### 2.3. Lipophilicity and Solubility Study

Lipophilicity refers to the ability of a molecule to dissolve in lipids and is measured by the distribution behavior in organic/aqueous systems using logP [37,38]. In terms of the antibacterial activity, lipophilicity plays a crucial role, as it affects the ability of a molecule to penetrate through the bacterial cell wall and exert its effects on the bacteria [39]. This is especially true for QACs, as their main mechanism includes membrane disruption [40,41].

A molecule with high lipophilicity can easily penetrate the bacterial cell membrane, which is composed primarily of lipids. This property renders lipophilic compounds more effective against Gram-negative bacteria, as they have an outer lipid layer which makes it difficult for hydrophilic (water-soluble) molecules to penetrate [42].

On the other hand, excessive lipophilicity can cause the cut-off effect that is described above [33]. Therefore, the lipophilicity of a molecule affects its ability to reach its target site in the bacteria and exert its antibacterial activity. The ideal lipophilicity of an antimicrobial agent depends on the type of microorganism and the site of infection.

It can be argued that a rapid decrease in activity is correlated with the lipophilicity drop in compounds possessing more oxygen atoms in the spacer chain. However, this theory is a subject of debate. For example, in the study of QACs based on pyridoxine and fatty acids, the authors found that all the compounds with logP < 0 were inactive on *S. aureus* [43]. Nevertheless, our compounds **7b**–**e** and **8a**–**b** showed good bacteriostatic and bactericidal effects against *S. aureus* in spite of the low lipophilicity, with logP below zero. Haldar et al. highlighted the utmost importance of maintaining the hydrophilic–hydrophobic balance of QACs [44]. This optimal balance enables a better interaction with the bacterial membrane through electrostatic interaction (hydrophilic) and membrane permeation (lipophilic), leading to a more efficient biocide action. Dong et al. stated that the ratio of cationic groups to hydrophobic groups is also the key to successful biofilm eradication [45].

Our lipophilicity assay supports the mentioned claims and provides some new insights (Figure 4). The optimal range of tris-QACs lipophilicity, when exposed to planktonic forms of bacterial cells, was, on average, in the range from −3.2 to −1.2 log P, except for *S. aureus*, for which this range extended much further (Figure 4A). For biofilms, the described range was narrower: from −3.2 to −2.2 (Figure 4B). Moreover, this trend was observed for all tested strains. Therefore, in order to achieve the best antibiofilm effect, maintaining the hydrophilic–hydrophobic balance is a more important parameter than the eradication of planktonic bacteria. However, this parameter is unique for each pathogen and requires further study.

Water solubility is also one of the most important and necessary features for good antibacterial and antibiofilm properties. The wettability test illustrated that the surface covered with biofilms possessed hydrophilic properties [47]. Therefore, the outer biofilm layer was hydrophilic and better permeated by water-soluble antibacterial agents. That was indeed the case with the tris-QACs series, where the more soluble compound **7c** possessed significantly higher antibiofilm activity than **7d**—the lead compound against the planktonic cells (Figure 5).

## 3. Materials and Methods

### 3.1. General Information

Reagents were purchased and used without further purification. All melting points were measured with a Stuart SMP30 melting point apparatus and were uncorrected. Melting point measurements were taken after product recrystallization and additional drying under reduced pressure. ^1^H and ^13^C NMR spectra were recorded with a Bruker AM300 at ambient temperature in DMSO-d_6_ or CDCl_3_ solutions. Chemical shift values are given in δ scale relative to Me_4_Si. The J values are given in hertz. Only discrete or characteristic signals for the ^1^H NMR are reported. IR spectra were recorded with a Bruker ALPHA-T FT-IR spectrometer in KBr pellets. HR-ESI-MS were measured on a Bruker microTOF II instrument (Bruker Daltonik GmbH, Bremen, Germany); external or internal calibration was performed with electrospray calibrant solution (Fluka). All starting materials were obtained from commercial sources and used without purification. All reactions were monitored with thin-layer chromatography (TLC) and carried out with Merck precoated plates DC-AlufolienKieselgel60 F254.

4-Alkylaminopyridines were obtained in three stages according to known methods [48,49].

#### 3.1.1. General Procedure for the Preparation of Ethylene Glycoles Dichoride **2a–c**

Ethylene glycol (50.51 mmol), triethylamine (50.51 mmol), and chloroform (8 mL) were placed in a three-necked round bottom flask of 100 mL, equipped with a drop funnel, a magnetic stirrer, and a reflux condenser. The solution was cooled to 5 °C in an ice bath. Thionyl chloride (126.28 mmol) was added dropwise while stirring into the solution, so that the temperature did not exceed 20 °C. Then, the mixture was heated under reflux for 8 h. The flask was cooled to room temperature. The reaction medium was sequentially treated with ml of 5% HCl, then H_2_O. The organic layer was separated, dried over anhydrous sodium sulfate, and evaporated. The residue was purified by vacuum distillation to provide a product in the form of a colorless liquid. The total yield was 80–83%.

**1-chloro-2-(2-chloroethoxy)ethane (2a)** Yield: 5.9 g (82%); colorless liquid; b.p. 70–71 °C at 19 Torr; 1H NMR (CDCl_3_, 300.13 MHz): δ 3.62 (t, J = 5.7 Hz, 4H, -CH_2_O), 3.75 (t, J = 5.8 Hz, 4H, -CH_2_Cl).

**1-chloro-2-(2-(2-(2-chloroethoxy)ethoxy)ethoxy)ethane (2b)** Yield: 7.84 g (83%); colorless liquid; b.p. 75–76 °C at 1.5 Torr; 1H NMR (CDCl_3_, 300.13 MHz): δ 3.65 (t, J = 5.7 Hz, 4H, CH_2_O), 3.71 (s, 4H, CH_2_O), 3.79 (t, J = 5.9 Hz, 4H, CH_2_Cl).

**1,2-bis(2-chloroethoxy)ethane (2c)** Yield: 9.33 g (80%); light-yellow liquid; b.p. 115–116 °C at 1.15 Torr; 1H NMR (CDCl_3_, 300.13 MHz): δ 3.63 (t, J = 5.8 Hz, 4H, CH_2_O), 3.67 (s, 8H, CH_2_O), 3.76 (t, J = 5.9 Hz, 4H, CH_2_Cl).

#### 3.1.2. General Procedure for the Preparation of **3a–c**

Isocyanuric acid (10 mmol) was added to a suspension of 60% sodium hydride (33 mmol) in hexamethylphosphoramide (HMPA) (15 mL). The mixture was stirred at a temperature of 100 °C for 3 h. After that, ethylene glycol dichlorides **2a–c** (40 mmol) was added dropwise, and heating was continued for 20 h. The reaction was then cooled to room temperature, and a small amount of distilled water was added to remove the NaH residues. The solvent was evaporated in a vacuum. The residue was purified by flash chromatography (eluent MeOH:chloroform = 1:100) to obtain a product in the form of a yellow oil. The overall yield was 51–60%.

**1,3,5-tris(2-(2-chloroethoxy)ethyl)-1,3,5-triazinane-2,4,6-trion (3a)** Yield: 2.69 g (60%); yellow oil; ^1^H NMR (CDCl_3_, 300.13 MHz): δ 3.58 (t, J = 5.6 Hz, 6H, CH_2_N), 3.69–3.79 (m, 12H, CH_2_O), 4.12 (t, J = 5.6 Hz, 6H, CH_2_Cl); ^13^C NMR (CDCl_3_, 75.47 MHz): δ 41.7, 42.9, 67.4, 70.7, 149.0. HRMS (ESI) *m*/*z* [M + H ]+ calcd for C_15_H_24_Cl_3_N_3_O_6_+: 448.0803; found: 448.0794.

**1,3,5-tris(2-(2-(2-chloroethoxy)ethoxy)ethyl)-1,3,5-triazinane-2,4,6-trione (3b)** Yield: 3.25 g (56%); light-yellow oil; ^1^H NMR (CDCl_3_, 300.13 MHz): δ 3.56–68 (m, 12H, CH_2_N), 3.70–3.80 (m, 12H, CH_2_O), 3.80–3.89 (m, 6H, CH_2_O), 4.11 (t, J = 5.8 Hz, 6H, CH_2_Cl); ^13^C NMR (CDCl_3_, 75.47 MHz): δ 41.7, 42.7, 67.5, 70.0, 70.6, 71.3, 149.0. HRMS (ESI) *m*/*z* [M + H]+ calcd for C21H36Cl3N3O9+: 580.1590; found: 580.1584.

**1,3,5-tris(2-(2-(2-(2-chloroethoxy)ethoxy)ethoxy)ethyl)-1,3,5-triazinane-2,4,6-trione (2c)** Yield: 3.63 g (51%); light-yellow oil; ^1^H NMR (CDCl_3_, 300.13 MHz): δ 3.62 (t, J = 5.6 Hz, 6H, CH_2_N), 3.71 (s, 24H, CH_2_O), 3.79–3.87 (m, 12H, CH_2_O), 4.11 (t, J = 5.9 Hz, 6H, CH_2_Cl); ^13^C NMR (CDCl_3_, 75.47 MHz): δ 41.7, 42.7, 62.4, 70.0, 70.5, 70.6 (s, 2C), 71.3, 149.0. HRMS (ESI) *m*/*z* [M + H]+ calcd for C27H48Cl3N3O12+: 712.2376; found: 712.2371.

The images of 1H, 13C NMR and HRMS spectra are represented in Appendix A.

#### 3.1.3. General Procedure for the Preparation of Novel Pyridinium Tris-QACs **7a–g**, **8a–c**, and **9a–c**

4-alkylaminopyridine **6a–g** (4 mmol) was added to a solution of **3a–c** (1.2 mmol) in 1-butanol (10 mL). The mixture was heated under reflux for 48 h, the reaction was cooled to room temperature, and the solvent was evaporated under vacuum. The crude residue was washed with acetone or diethyl ether (20–25 mL), and the precipitate was dried until a white solid product was obtained. The total yield was 66–83%. For long-term storage, the substances were placed in a desiccator over phosphoric anhydride.


**1,1′,1″-((((2,4,6-trioxo-1,3,5-triazinane-1,3,5-triyl)tris(ethane-2,1-diyl))tris(oxy))tris(ethane-2,1-diyl))tris(4-(hexylamino)pyridin-1-ium) trichloride (7a)**


Yield: 0.89 g (76%); White hygroscopic powder; m.p.: 64–67 °C; IR (KBr): ν 3370 (NH), 2923 (N^+^), 2853, 2069, 1692 (C=O), 1587, 1466(CH), 1349 (C-N_ar_), 1111 (C-O), 1058 (C-N_amine_); ^1^H NMR (DMSO-d_6_, 300.13 MHz): δ 0.88 (t, J = 6.2 Hz, 9H, CH_3_), 1.24–1.42 (m, 18H, CH_2_), 1.50–1.63 (m, 6H, CH_2_), 3.24 (t, *J* = 6.9 Hz, 6H, CH_2_NH), 3.55 (t, J = 5.3 Hz, 6H, CH_2_N), 3.67–3.79 (m, 6H, CH_2_O), 3.86 (t, J = 5.1 Hz, 6H, CH_2_O), 4.20–4.36 (m, 6H, CH_2_N+), 6.84–6.92 (m, 3H, CH_ar_), 6.94–7.04 (m, 3H, CH_ar_), 8.05 (d, J = 7.1 Hz, 3H, CH_ar_), 8.23 (d, J = 7.3 Hz, 3H, CH_ar_), 9.14 (t, *J* = 5.4 Hz, 1H, NH); ^13^C NMR (DMSO-d_6_, 75.47 MHz): δ 14.3, 22.5, 26.4, 28.3, 31.3, 41.7, 42.6, 42.7, 56.9, 67.1, 69.0, 105.2, 110.3, 142.0, 144.4, 149.0, 157.2. HRMS (ESI) *m*/*z* [M + H − 3Cl]+ calcd for C48H75N9O6+: 874.5913; found: 874.5895.


**1,1′,1″-((((2,4,6-trioxo-1,3,5-triazinane-1,3,5-triyl)tris(ethane-2,1-diyl))tris(oxy))tris(ethane-2,1-diyl))tris(4-(heptylamino)pyridin-1-ium) trichloride (7b)**


Yield: 0.92 g (75%); White hygroscopic powder; m.p.: 72–75 °C; IR (KBr): ν 3433 (NH), 2924 (N^+^), 2853, 2068, 1692 (C=O), 1587, 1466(CH), 1349 (C-N_ar_), 1112 (C-O), 1059 (C-N_amine_); ^1^H NMR (DMSO-d_6_, 300.13 MHz): δ 0.87 (t, J = 6.2 Hz, 9H, CH_3_), 1.17–1.39 (m, 24H, CH_2_), 1.46–1.64 (m, 6H, CH_2_), 3.24 (t, *J* = 6.7 Hz, 6H, CH_2_NH), 3.55 (t, J = 4.8 Hz, 6H, CH_2_N), 3.79–3.68 (m, 6H, CH_2_O), 3.86 (t, J = 4.5 Hz, 6H, CH_2_O), 4.39–4.21 (m, 6H, CH_2_N+), 6.91–6.82 (m, 3H, CH_ar_), 7.07–6.94 (m, 3H, CH_ar_), 8.06 (d, J = 7.2 Hz, 3H, CH_ar_), 8.23 (d, J = 7.1 Hz, 3H, CH_ar_), 9.19 (t, J = 5.2 Hz, 1H, NH); ^13^C NMR (DMSO-d_6_, 75.47 MHz): δ 14.4, 22.5, 26.8, 28.4, 28.8, 31.6, 41.7, 42.6, 42.7, 56.9, 67.1, 69.0, 105.2, 110.3, 142.04, 144.42, 149.0, 157.2. HRMS (ESI) *m*/*z* [M + H − 3Cl]+calcd for C51H81N9O6+: 916.6383; found: 916.6409.


**1,1′,1″-((((2,4,6-trioxo-1,3,5-triazinane-1,3,5-triyl)tris(ethane-2,1-diyl))tris(oxy))tris(ethane-2,1-diyl))tris(4-(octylamino)pyridin-1-ium) trichloride (7c)**


Yield: 1.01 g (79%); White hygroscopic powder; m.p.: 83–86 °C; IR (KBr): ν 3414 (NH), 2924 (N^+^), 2852, 2069, 1692 (C=O), 1587, 1466(CH), 1350 (C-N_ar_), 1113 (C-O), 1059 (C-N_amine_); ^1^H NMR (DMSO-d_6_, 300.13 MHz) δ 0.87 (t, J = 6.2 Hz, 9H, CH_3_), 1.16–1.38 (m, 30H, CH_2_), 1.46–1.63 (m, 6H, CH_2_), 3.24 (t, *J* = 6.8 Hz, 6H, CH_2_NH), 3.55 (t, J = 5.4 Hz, 6H, CH_2_N), 3.66–3.79 (m, 6H, CH_2_O), 3.86 (t, J = 5.1 Hz, 6H, CH_2_O), 4.19–4.38 (m, 6H, CH_2_N+), 6.87–6.94 (m, 3H, CH_ar_), 6.95–7.06 (m, 3H, CH_ar_), 8.05 (d, *J* = 7.1 Hz, 3H, CH_ar_), 8.23 (d, J = 7.3 Hz, 3H, CH_ar_), 9.15 (t, J = 5.4 Hz, 1H, NH); ^13^C NMR (DMSO-d_6_, 75.47 MHz): δ 14.4, 22.5, 26.8, 28.4, 29.0, 29.1, 31.7, 41.7, 42.6, 42.7, 56.9, 67.1, 69.0, 105.2, 110.3, 142.0, 144.4, 149.0, 157.2. HRMS (ESI) *m*/*z* [M + H − 3Cl]+calcd for C54H87N9O6^2+^: 479.8462; found: 479.8481.


**1,1′,1″-((((2,4,6-trioxo-1,3,5-triazinane-1,3,5-triyl)tris(ethane-2,1-diyl))tris(oxy))tris(ethane-2,1-diyl))tris(4-(nonylamino)pyridin-1-ium) trichloride (7d)**


Yield: 1.01 g (76%); White powder; m.p.: 91–93 °C; IR (KBr): ν 3436 (NH), 2922 (N^+^), 2853, 2069, 1691 (C=O), 1587, 1466(CH), 1348 (C-N_ar_), 1111 (C-O), 1058 (C-N_amine_); ^1^H NMR (DMSO-d_6_, 300.13 MHz): δ 0.85 (t, J = 6.2 Hz, 9H, CH_3_), 1.16–1.43 (m, 36H, CH_2_), 1.47–1.65 (m, 6H, CH_2_), 3.22 (t, *J* = 7.0 Hz, 6H, CH_2_NH), 3.54 (t, J = 5.6 Hz, 6H, CH_2_N), 3.73 (t, J = 4.3 Hz, 6H, CH_2_O), 3.85 (t, J = 5.5 Hz, 6H, CH_2_O), 4.18–4.39 (m, 6H, CH_2_N+), 6.81–6.90 (m, 3H, CH_ar_), 6.92–7.05 (m, 3H, CH_ar_), 8.04 (d, J = 8.7 Hz, 3H, CH_ar_), 8.22 (d, J = 7.4 Hz, 3H, CH_ar_), 9.15 (t, J = 6.1 Hz, 1H, NH); ^13^C NMR (DMSO-d_6_, 75.47 MHz): δ 14.4, 22.5, 26.8, 28.4, 29.1, 29.2, 29.4, 31.7, 41.7, 42.6, 42.7, 56.9, 67.2, 69.0, 105.2, 110.3, 142.0, 144.4, 149.0, 157.2. HRMS (ESI) *m*/*z* [M + H − 3Cl]+calcd for C57H93N9O6+: 1000.7322; found: 1000.7295.


**1,1′,1″-((((2,4,6-trioxo-1,3,5-triazinane-1,3,5-triyl)tris(ethane-2,1-diyl))tris(oxy))tris(ethane-2,1-diyl))tris(4-(decylamino)pyridin-1-ium) trichloride (7e)**


Yield: 1.11 g (80%); White powder; m.p.: 102–104 °C; IR (KBr): ν 3385 (NH), 2922 (N^+^), 2852, 2070, 1692 (C=O), 1587, 1467(CH), 1349 (C-N_ar_), 1111 (C-O), 1058 (C-N_amine_); ^1^H NMR (DMSO-d_6_, 300.13 MHz): δ 0.85 (t, J = 6.6 Hz, 9H, CH3), 1.16–1.39 (m, 42H, CH_2_), 1.46–1.63 (m, 6H, CH_2_), 3.22 (t, *J* = 6.9 Hz, 6H, CH_2_NH), 3.54 (t, *J* = 5.5 Hz, 6H, CH_2_N), 3.69–3.78 (m, J = 6.3 Hz, 6H, CH_2_O), 3.85 (t, J = 5.4 Hz, 6H, CH_2_O), 4.21–4.33 (m, 6H, CH_2_N+), 6.83–6.88 (m, 3H, CH_ar_), 6.92–7.03 (m, 3H, CH_ar_), 8.04 (d, J = 6.4 Hz, 3H, CH_ar_), 8.22 (d, J = 7.3 Hz, 3H, CH_ar_), 9.05–9.21 (m, 1H, NH); 13C NMR (DMSO-d_6_, 75.47 MHz): δ 14.4, 22.6, 26.8, 28.4, 29.1, 29.4(2C), 29.5, 31.8, 41.7, 42.7, 56.9, 67.1, 69.1, 105.2, 110.3, 142.1, 144.4, 149.1, 157.2. HRMS (ESI) *m*/*z* [M + H − 3Cl]+calcd for C60H102N9O6^+^: 1176.8741; found: 1176.8793.


**1,1′,1″-((((2,4,6-trioxo-1,3,5-triazinane-1,3,5-triyl)tris(ethane-2,1-diyl))tris(oxy))tris(ethane-2,1-diyl))tris(4-(undecylamino)pyridin-1-ium)trichloride (7f)**


Yield: 1.16 g (81%); White powder; m.p.: 112–116 °C; IR (KBr): ν 3396 (NH), 2923 (N^+^), 2853, 2069, 1692 (C=O), 1587, 1466(CH), 1349 (C-N_ar_), 1112 (C-O), 1058 (C-N_amine_); ^1^H NMR (DMSO-d_6_, 300.13 MHz): δ 0.85 (t, J = 6.5 Hz, 9H, CH_3_), 1.16–1.4 (m, 48H, CH_2_), 1.46–1.63 (m, 6H, CH_2_), 3.22 (t, J = 6.8 Hz, 6H, CH_2_NH), 3.54 (t, J = 5.1 Hz, 6H, CH_2_N), 3.65–3.78 (m, 6H, CH_2_O), 3.85 (t, J = 5.1 Hz, 6H, CH_2_O), 4.07–4.50 (m, 6H, CH_2_N+), 6.73–6.93 (m, 3H, CH_ar_), 6.92–7.13 (m, 3H, CH_ar_), 8.04 (d, J = 6.5 Hz, 3H, CH_ar_), 8.21 (d, J = 6.7 Hz, 3H, CH_ar_), 9.05–9.20 (m, 1H, NH); ^13^C NMR (DMSO-d_6_, 75.47 MHz): δ 14.4, 22.5, 26.8, 28.3, 29.2 (s, 4C), 29.5, 31.7, 41.7, 42.6, 42.7, 56.9, 67.2, 69.0, 105.2, 110.3, 142.1, 144.4, 149.1, 157.1. HRMS (ESI) *m*/*z* [M + H − 3Cl]+calcd for C63H105N9O6^2+^: 542.9167; found: 542.9193.


**1,1′,1″-((((2,4,6-trioxo-1,3,5-triazinane-1,3,5-triyl)tris(ethane-2,1-diyl))tris(oxy))tris(ethane-2,1-diyl))tris(4-(dodecylamino)pyridin-1-ium)trichloride (7g)**


Yield: 1.23 g (83%); White powder; m.p.: 123–127°C; IR (KBr): ν 3370 (NH), 2923 (N^+^), 2853, 2069, 1692 (C=O), 1587, 1469(CH), 1349 (C-N_ar_), 1112 (C-O), 1060 (C-N_amine_); ^1^H NMR (DMSO-d_6_, 300.13 MHz): δ 0.84 (t, J = 6.3 Hz, 9H, CH_3_), 1.16–1.39 (m, 54H, CH_2_), 1.46–1.63 (m, 6H, CH_2_), 3.21 (t, J = 6.5 Hz, 6H, CH_2_NH), 3.49–3.58 (m, 6H, CH_2_N), 3.68–3.78 (m, 6H, CH_2_O), 3.80–3.90 (m, 6H, CH_2_O), 4.21–4.38 (m, 6H, CH_2_N+), 6.81–6.89 (m, 3H, CH_ar_), 7.00–7.11 (m, 3H, CH_ar_), 8.07 (d, J = 7.2 Hz, 3H, CH_ar_), 8.25 (d, J = 7.4 Hz, 3H, CH_ar_), 9.05–9.28 (m, 1H, NH); ^13^C NMR (DMSO-d_6_, 75.47 MHz): δ 14.4, 22.5, 26.8, 28.4, 29.2 (s, 5C), 29.5, 31.8, 41.7, 42.6, 42.7, 56.8, 67.2, 69.0, 105.1, 110.2, 142.0, 144.4, 149.0, 157.1. HRMS (ESI) *m*/*z* [M + H − 3Cl]+ calcd for C66H111N9O6+: 1126.8730; found: 1126.8746.

The structure was confirmed by 2D 1H-13C HSQC, 1H-13C HMBC and 1H-1H COSY spectra. The key interactions are represented in Appendix A.


**1,1′,1″-((((((2,4,6-trioxo-1,3,5-triazinane-1,3,5-triyl)tris(ethane-2,1-diyl))tris(oxy))tris(ethane-2,1-diyl))tris(oxy))tris(ethane-2,1-diyl))tris(4-(octylamino)pyridin-1-ium) trichloride (8a)**


Yield: 1.00 g (76%); White hygroscopic powder; m.p.: 72–75°C; IR (KBr): ν 3452 (NH), 2923 (N^+^), 2853, 2069, 1690 (C=O), 1587, 1466(CH), 1349 (C-N_ar_), 1110 (C-O), 1058 (C-N_amine_); ^1^H NMR (DMSO-d_6_, 300.13 MHz): δ 0.85 (t, J = 6.3 Hz, 9H, CH_3_), 1.17–1.41 (m, 30H, CH_2_), 1.54–1.64 (m, 6H, CH_2_), 3.18–3.39 (m, 6H, CH_2_NH), 3.44–3.59 (m, 18H, CH_2_O), 3.67–3.78 (m, 6H, CH_2_N), 3.89 (t, J = 5.5 Hz, 6H, CH_2_O), 4.22–4.36 (m, 6H, CH_2_N+), 6.87–6.95 (m, 3H, CH_ar_), 6.99–7.10 (m, 3H, CH_ar_), 8.10 (d, J = 7.1 Hz, 3H, CH_ar_), 8.26 (d, J = 7.2 Hz, 3H, CH_ar_) 9.25 (br.s, 3H, NH); ^13^C-NMR (DMSO-d_6_, 75.47 MHz): δ 14.0, 22.5, 26.8, 28.4, 29.3(2C), 31.7, 41.7, 42.7, 56.7, 67.1, 69.5, 69.7, 70.0, 105.3, 110.5, 142.0, 144.5, 149.5, 157.2. HRMS (ESI) *m*/*z* [M + H − 3Cl]+calcd for C60H99N9O9+: 1090.7639; found: 1090.7611.


**1,1′,1″-((((((2,4,6-trioxo-1,3,5-triazinane-1,3,5-triyl)tris(ethane-2,1-diyl))tris(oxy))tris(ethane-2,1-diyl))tris(oxy))tris(ethane-2,1-diyl))tris(4-(decylamino)pyridin-1-ium) trichloride (8b)**


Yield: 1.87 g (77%); White hygroscopic powder; m.p.: 91–95°C; IR (KBr): ν 3373 (NH), 2921 (N^+^), 2853, 2069, 1693 (C=O), 1587, 1466(CH), 1349 (C-N_ar_), 1112 (C-O), 1058 (C-N_amine_); ^1^H NMR (DMSO-d_6_, 300.13 MHz): δ 0.85 (t, J = 6.1 Hz, 9H, CH_3_), 1.16–1.42 (m, 42H, CH_2_), 1.47–1.62 (m, 6H, CH_2_), 3.18–3.30 (m, 6H, CH_2_NH), 3.44–3.60 (m, 18H, CH_2_O), 3.66–3.79 (m, 6H, CH_2_N), 3.89 (t, J = 5.5 Hz, 6H, CH_2_O), 4.37–4.42 (m, 6H, CH_2_N+), 6.87–6.96 (m, 3H, CH_ar_), 7.01–7.11 (m, 3H, CH_ar_), 8.11 (d, J = 7.2 Hz, 3H, CH_ar_), 8.26 (d, J = 7.0 Hz, 3H, CH_ar_) 9.25 (br.s, 3H, NH); ^13^C-NMR (DMSO-d_6_, 75.47 MHz): δ 14.4, 22.5, 26.8, 28.4, 29.3(2C), 31.7, 41.7, 42.7, 56.7, 67.1, 69.6, 69.8, 70.0, 105.3, 110.4, 142.0, 144.4, 149.2, 157.2. HRMS (ESI) *m*/*z* [M + H − 3Cl]+calcd for C66H112N9O9^+^: 1174.8578; found: 1174.8582.


**1,1′,1″-((((((2,4,6-trioxo-1,3,5-triazinane-1,3,5-triyl)tris(ethane-2,1-diyl))tris(oxy))tris(ethane-2,1-diyl))tris(oxy))tris(ethane-2,1-diyl))tris(4-(dodecylamino)pyridin-1-ium) trichloride (8c)**


Yield: 1.20 g (73%); White hygroscopic powder; m.p.: 112–114°C; IR (KBr): ν 3464 (NH), 2923 (N^+^), 2853, 2069, 1692 (C=O), 1587, 1466(CH), 1349 (C-N_ar_), 1111 (C-O), 1058 (C-N_amine_); ^1^H NMR (DMSO-d_6_, 300.13 MHz): δ 0.85 (t, J = 6.3 Hz, 9H, CH_3_), 1.15–1.39 (m, 54H, CH_2_), 1.48–1.62 (m, 6H, CH_2_), 3.17–3.29 (m, 6H, CH_2_NH), 3.43–3.59 (m, 18H, CH_2_O), 3.66–3.76 (m, 6H, CH_2_N), 3.88 (t, J = 5.1 Hz, 6H, CH_2_O), 4.22–4.35 (m, 6H, CH_2_N+), 6.85–6.91 (m, 3H, CH_ar_), 6.97–7.07 (m, 3H, CH_ar_), 8.09 (d, J = 7.7 Hz, 3H, CH_ar_), 8.25 (d, J = 7.0 Hz, 3H, CH_ar_) 9.13 (br.s, 3H, NH); ^13^C-NMR (DMSO-d_6_, 75.47 MHz): δ 14.4, 22.5, 26.8, 28.4, 29.3(3C), 31.8, 41.7, 42.7, 56.8, 67.1, 69.5, 69.8, 70.0, 105.3, 110.5, 142.0, 144.5, 149.2, 157.21. HRMS (ESI) *m*/*z* [M + H − 3Cl]+calcd for C72H123N9O9^+^: 1258.9517; found: 1258.9525.


**1,1′,1″-((((((((2,4,6-trioxo-1,3,5-triazinane-1,3,5-triyl)tris(ethane-2,1-diyl))tris(oxy))tris(ethane-2,1-diyl))tris(oxy))tris(ethane-2,1-diyl))tris(oxy))tris(ethane-2,1-diyl))tris(4-(octylamino)pyridin-1-ium) trichloride (9a)**


Yield: 1.12 g (70%); White hygroscopic powder; m.p.: 59–64 °C; IR (KBr): ν 3377 (NH), 2923 (N^+^), 2851, 2069, 1692 (C=O), 1587, 1466(CH), 1349 (C-N_ar_), 1112 (C-O), 1058 (C-N_amine_); ^1^H NMR (DMSO-d_6_, 300.13 MHz): δ 0.86 (t, J = 6.4 Hz, 9H, CH_3_), 1.19–1.40 (m, 30H, CH_2_), 1.50–1.62 (m, 6H, CH_2_), 3.20–3.29 (m, 6H, CH_2_N), 3.41–3.58 (m, 30H, CH_2_O), 3.74 (t, J = 4.4 Hz, 6H, CH_2_N), 3.91 (t, J = 5.9 Hz, 6H, CH_2_N), 4.29(t, J = 4.2 Hz 6H, CH_2_N+), 6.87–7.02 (m, 6H, CH_ar_), 8.08 (d, J = 7.2 Hz, 3H, CH_ar_), 8.24 (d, J = 7.1 Hz, 3H, CH_ar_) 9.00 (br.s, 3H, NH); ^13^C-NMR (DMSO-d_6_, 75.47 MHz): δ 14.4, 22.5, 26.8, 28.4 (2C), 29.1, 31.7, 41.6, 42.7, 56.8, 67.0, 69.5, 69.9, 70.1, 70.2, 79.7, 105.3, 110.5, 142.1, 144.5, 149.2, 157.2. HRMS (ESI) *m*/*z* [M + H − 3Cl]+calcd for C66H111N9O12^+^: 1222.8425; found: 1222.8411.


**1,1′,1″-((((((((2,4,6-trioxo-1,3,5-triazinane-1,3,5-triyl)tris(ethane-2,1-diyl))tris(oxy))tris(ethane-2,1-diyl))tris(oxy))tris(ethane-2,1-diyl))tris(oxy))tris(ethane-2,1-diyl))tris(4-(decylamino)pyridin-1-ium) trichloride (9b)**


Yield: 1.14 g (67%); White hygroscopic powder; m.p.: 75–78 °C; IR (KBr): ν 3371 (NH), 2923 (N^+^), 2853, 2069, 1692 (C=O), 1587, 1466(CH), 1349 (C-N_ar_), 1111 (C-O), 1060 (C-N_amine_); ^1^H NMR (DMSO-d_6_, 300.13 MHz): δ 0.85 (t, J = 6.3 Hz, 9H, CH_3_), 1.20–1.39 (m, 42H, CH_2_), 1.48–1.64 (m, 6H, CH_2_), 3.19–3.30 (m, 6H, CH_2_N), 3.41–3.58 (m, 30H, CH_2_O), 3.69–3.78 (m, 6H, CH_2_N), 3.83–3.96 (m, 6H, CH_2_N), 4.25–4.34(m, 6H, CH_2_N+), 6.85–7.00 (m, 6H, CH_ar_), 8.08 (d, J = 7.2 Hz, 3H, CH_ar_), 8.24 (d, J = 6.8 Hz, 3H, CH_ar_) 8.94 (br.s, 3H, NH); ^13^C NMR (DMSO-d_6_, 75.47 MHz): δ 14.4, 22.5, 26.8, 28.4, 29.1 (2C), 29.4, 29.4, 31.7, 42.7, 42.8, 56.9, 67.0, 69.5, 69.9, 70.0, 70.1, 70.2, 105.3, 110.5, 142.2, 144.5, 149.2, 157.2. HRMS (ESI) *m*/*z* [M + H − 3Cl]+calcd for C72H123N9O12^+^: 1306.9364; found: 1306.9355.


**1,1′,1″-((((((((2,4,6-trioxo-1,3,5-triazinane-1,3,5-triyl)tris(ethane-2,1-diyl))tris(oxy))tris(ethane-2,1-diyl))tris(oxy))tris(ethane-2,1-diyl))tris(oxy))tris(ethane-2,1-diyl))tris(4-(dodecylamino)pyridin-1-ium) trichloride (9c)**


Yield: 1.19 g (66%); White hygroscopic powder; m.p.: 94–97 °C; IR (KBr): ν 3432 (NH), 2924 (N^+^), 2853, 2068, 1692 (C=O), 1587, 1466(CH), 1349 (C-N_ar_), 1111 (C-O), 1059 (C-N_amine_); ^1^H NMR (DMSO-d_6_, 300.13 MHz): δ 0.85 (t, J = 6.4 Hz, 9H, CH_3_), 1.11–1.43 (m, 54H, CH_2_), 1.49–1.62 (m, 6H, CH_2_), 3.19–3.30 (m, 6H, CH_2_N), 3.41–3.58 (m, 30H, CH_2_O), 3.69–3.78 (m, 6H, CH_2_N), 3.83–3.96 (m, 6H, CH_2_N), 4.28–4.34(m, 6H, CH_2_N+), 6.86–7.02 (m, 6H, CH_ar_), 8.09 (d, J = 7.3 Hz, 3H, CH_ar_), 8.24 (d, J = 7.0 Hz, 3H, CH_ar_) 9.02 (br.s, 3H, NH); ^13^C-NMR (DMSO-d_6_, 75.47 MHz): δ 14.4, 22.6, 26.8, 28.4, 29.1 (3C), 29.5, 31.8, 41.7, 42.7, 56.8, 67.0, 69.5, 69.7, 69.9, 70.1, 70.2, 105.2, 110.5, 142.1, 144.5, 149.2, 157.2. HRMS (ESI) *m*/*z* [M + H − 3Cl]+calcd for C78H136N9O12^+^: 1391.0303; found: 1391.0296.

The images of 1H, 13C NMR and HRMS spectra are represented in Appendix A.

### 3.2. Bacterial Strains

Reference strains of microorganisms *E. coli* ATCC 25922, *K. pneumoniae* ATCC 70060, *S. aureus* ATCC 43300, *P. aeruginosa* ATCC 27853, and *A. baumannii* ATCC 15308 were obtained from the State Collection of Pathogenic Microorganisms (Obolensk, Russia). Clinical isolates of *E. coli* B-3421/19, *K. pneumoniae* B-2523/18, *S. aureus* B-8648, *P. aeruginosa* B-2099/18, and *A. baumannii* B-2926/18 were isolated in the Molecular Microbiology department of the Federal Budget Institution of Science State Research Center for Applied Biotechnology and Microbiology (FBSI SRC PMB, Obolensk, Russia) from clinical samples in the investigation of infection cases in 2016–2018.

### 3.3. Identification of Microorganisms

Species identification of microorganisms was carried out on an MALDI-TOF Biotyper mass spectrometer (Bruker, Germany). One part of the daily bacterial culture was introduced into a 2 mL tube (Eppendorf, Hamburg, Germany) containing 300 µL of deionized water and was triturated until a homogeneous suspension was obtained. Then, 900 µL of 96% ethanol was added and shaken. The tube was then centrifuged at 12,000× *g* for 2 min. The supernatant was discarded, and the precipitate was dried in air. After drying, the precipitate was thoroughly mixed with 30 µL of 70% formic acid, incubated for 10 min at room temperature, then mixed with 30 µL of acetonitrile (Sigma-Aldrich, St. Louis, MO, USA) and incubated for another 10 min. This mixture was centrifuged at 12,000× *g* for 2 min, and then 1 µL of the supernatant was transferred to an MSP 96 Polished Steel MALDI Target Plate (Bruker Daltonik GmbH, Bremen, Germany), air-dried, and covered with 1 µL of a saturated solution of α-cyano-4-hydroxycinnamic acid (Bruker Daltonik, Bremen, Germany). The MALDI target plate was placed in an MALDI-TOF microflex LRF instrument (Bruker Daltonik, Bremen, Germany). The data acquisition settings were as follows: ion source 1 at 19.50 kV, ion source 2 at 18.22 kV, lens at 7.01 kV, and a mass range of 2000 to 20,000 Da. Spectrum registration was performed automatically using the MALDI Biotyper RTC 3.1 software (Bruker Daltonik, Bremen, Germany). The obtained spectra were compared with the reference spectra of the FFL v1.0 database. Identification scores ≥ 2.0 and 1.7–1.99 indicated recognition of the species and genus levels, respectively, while scores <1.7 indicated a lack of reliable data.

### 3.4. Antibacterials

All investigated tris-QACs (including TQAC-8) were synthesized at the N. D. Zelinsky Institute of Organic Chemistry Russian Academy of Sciences. Benzalkonium chloride was purchased from Acros Organics and used without further purification. Cetylpyridinium chloride was synthesized from pyridine using the simple method of alkylation in acetonitrile. Octenidine dihydrochloride was synthesized using a previously described method [50].

### 3.5. Cultivation of Microorganisms

Microorganisms were cultivated using agar and GRM broth. Cultivation of microorganisms was carried out for 20–24 h at a temperature of 37 °C.

### 3.6. Antibacterial Assay

The synthesized tris-QACs were dissolved in the DMSO:water system at a ratio of 1:9, with a final concentration of 1 mg/mL. For better dissolution, the samples were sonicated for 15 min at 40 °C. The MIC and MBC of the antimicrobials were determined by the method involving microdilution in culture broth, as indicated by the Clinical and Laboratory Standards Institute of the United States of America [51]. For this, twofold dilutions of the tested solutions (500–0.25 mg/L) were prepared in nutrient broth. The nutrient broth with the tested QACs (0.1 mL) was added to 12 wells in the horizontal rows of the culture plate. In separate rows, nutrient broth without QACs was added for control measures. In the case of anti-biofilm analysis, nutrient broth with the appropriate concentration of the tested compounds was added in 0.1 mL to 12 wells in horizontal rows of a culture plate with washed biofilms, which were prepared according to the abovementioned method. From single colonies grown on GRM medium at 37 °C for 18 h, a suspension was prepared with an optical density of 0.5 according to the McFarland standard in sterile saline, which corresponds to approximately 1–2 × 10^8^ CFU/mL. The suspension was then diluted 100-fold by adding 0.2 mL of the suspension to a flask containing 19.8 mL of Mueller–Hinton nutrient agar (MHA). The concentration of microorganisms, in this case, was 10^6^ CFU/mL. The amount of 0.1 mL of the initial suspension was added to the wells with the test drug and control wells with broth. The final concentration of the microorganisms in each well was 5 × 10^5^ CFU/mL. The plates were covered with lids and placed in a thermostat (37 °C) for 20 h. The presence of bacterial growth was taken into account visually (according to the presence of turbidity in the well). The MIC was taken to be the minimum concentration of the preparation at which bacterial growth was absent after 48 h of incubation (Appendix A). MBC was determined by the results of inoculation on dense nutrient media. To accomplish this, from all the wells in which there was no visible growth (according to the presence of turbidity), 10 µL was sown on MHA. The results took into account the presence of culture growth at the site of application after 48 h of incubation at 37 °C. If there was no growth in the well, but the growth of the studied culture was observed when seeding from this well on solid nutrient medium, this concentration was taken as the MIC. The lowest concentration was taken as MBC, at which cell growth was completely suppressed when seeded on a dense nutrient medium (Appendix A). Since the limit of detection for this technique was 10 cfu/mL, the absence of any growth on MHA plate indicated that the concentration lay below this value. The initial concentration of 10^5^ cfu/mL was, thus, reduced to below 10 cfu/mL. Consequently, the MBC was effectively deemed to be the minimum concentration of antimicrobial capable of inactivating more than 99.99% of the bacteria present. Three replicates were performed for each strain and antimicrobial compound.

### 3.7. Solubility in Water

Tris-QAC (100 mg) **TQAC-8** was added to 1.0 mL of distilled water. Tris-QACs **7a–g**, **8a–c**, and **9a–c** (500 mg) were added to 1.0 mL of distilled water. The mixture was stirred at room temperature until the dissolution reached the dissolution equilibrium. Insoluble solids were filtered through a Schott filter, washed with cold acetone, and then dried at 60 °C. for 24 h. Solubility in water of novel tris-QACs was calculated (Appendix A) by the formula [52]:Solubility in water = (m_1_ − m_2_)/1 
where m_1_ (mg) is the mass of solids, and m_2_ (mg) is the weight of the undissolved solids.

## 4. Conclusions

In conclusion, we demonstrated that the incorporation of the ether group in cyanuric tris-QACs resulted in novel compounds that efficiently inhibited and eradicated the planktonic cells and biofilms formed by several clinically important Gram-positive bacteria (*S. aureus* ATCC 43300 and *S. aureus* B-8648), Gram-negative bacteria (*E. coli* ATCC 25922, *K. pneumoniae* ATCC 70060, *P. aeruginosa* ATCC 27853, *A. baumannii* ATCC 15308, *E. coli* B-3421/19, *K. pneumoniae* B-2523/18, *P. aeruginosa* B-2099/18, and *A. baumannii* B-2926/18), and fungi (*C. albicans* ATCC 10,231 and *C. albicans* AV-85). The established synthetic approach afforded good overall product yields from the available resources as glycols and lipophilic acids.

It is known that the QAC mechanism of action on biofilms does not differ from that of the planktonic forms. However, significantly higher concentrations of substances are required to permeate the exopolysaccharide matrix. Here, we hypothesized that tuning the hydrophilic–hydrophobic balance of QACs can significantly improve their antibiofilm potential, making them a promising platform for combating mature biofilms. Thus, substance **7c** showed inferior bioactivity on the planktonic cells compared with the bis-cation analogue **OCT**, but it outperformed it and most other QACs in our library (>150) on biofilms. We hope that the presented work will provide useful insights and form the basis for future research on the topic.

## Data Availability

Not applicable.

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
