# Peer review of "The Antimicrobial and Antibiofilm Potential of New Water-Soluble Tris-Quaternary Ammonium Compounds"

_ijms, 2023, doi:10.3390/ijms241310512_

Round 1

Reviewer 1 Report

Comments; 1. All investigated tris-QACs were dissolved in DMSO>>> the concentration and amount of DMSO 2. The synthesized tris-QACs were evaluated as an antibacteral>>> The statistical analysis is missing ... I didn't see any real photos of the antibacterial activity 3. 106 CFU/mL. correct. 4. 5 × 105 CFU/mL correct. 5. 1–2 × 108 CFU/mL correct. 6. Peaks and charts from 1H NMR should be inserted in the manuscript.

Minor errors.

Author Response

Comments;

  1. All investigated tris-QACs were dissolved in DMSO>>> the concentration and amount of DMSO

Al tested QACs were dissolved in DMSO/water system (1/9) to achieve 1000 mg/L concentration.(section 3.6, page 14)

  1. The synthesized tris-QACs were evaluated as an antibacteral>>> The statistical analysis is missing ... I didn't see any real photos of the antibacterial activity

Statistical analysis of microbiological activity was presented in the form of tables with minimum inhibitory concentrations and minimum bactericidal concentrations values, as well as graphs of the dependence of microbiological activity on lipophilicity. This approach meets all the requirements for the presentation of microbiological studies and was used in many publications earlier. Real photos are not a mandatory part of the activity evaluation. Below are examples of articles that use the same methods for assessing antibacterial activity.

https://doi.org/10.3390/ph15050514

https://doi.org/10.1002/cmdc.202100284

https://doi.org/10.1021/acsmedchemlett.9b00379

https://doi.org/10.1021/ml500203p

  1. 106 CFU/mL. correct. 4. 5 × 105 CFU/mL correct. 5. 1–2 × 108 CFU/mL correct.

Corrected accordingly (p. 14)

  1. Peaks and charts from 1H NMR should be inserted in the manuscript.

1H NMR spectra were added in Supporting Information

We thank the Reviewer for the comments!

Reviewer 2 Report

The authors have presented a study on preparation of a series of trimeric pyridinium salts assembled on a cyanuric acid core and their evaluation as antimicrobial agents. This structural motif has been already studied in several publications from the same group of authors. All synthetic procedures employed are quite standard and no significant improvent of bioactivity was achieved. Not only originality and novelty of this work is questionable but also the quality of the provided experimental data. My biggest concern is that no copies of NMR data were provided to support the purity and structure of the target compounds and moreover any data supporting their chemical formula such as HRMS or elemental analysis is missing. Unfortunately, I cannot recommend publication of this manuscript in its present form.

Additional comments and questions:

1. Copies of 1H and 13NMR spectra of all novel compounds must be provided (in a separated ESI file) to support their purity and structure. 

2. HRMS or elemental analysis data must be provided for all novel compounds to support their chemical formula

3. Schemes in the main text should be provided with better resolution

4. Yields should be added to reaction schemes. It is quite difficult to follow the discussion of yields while they are reported only in the materials and methods section.

5. NMR data should be double-checked. For example, it is unclear how can compound with 48 carbons (7a), which is symmetrical trimer with two identical carbons in pyridinium moiety has 18 signals in 13 NMR spectrum report.

6. The authors describe target compound as hygroscopic. How were they dried prior yield and melting point determination? How were they stored?

7. This part of general information is unclear: “Only discrete or characteristic signals for the 1H NMR are reported.” Partial analytical data should not be used for characterization of novel compounds.

8. On p.14 a procedure for water solubility determination is reported, but not the results were not discussed in the main text

9. LogP can determined experimentally quite easily, especially for gram-scale amounts of substances. Why did the authors use calculated values instead? 

10. Extensive editing of English language required. For example, term “reverse refrigerator” was used instead of “reflux condenser”

Extensive editing of English language required. For example, term “reverse refrigerator” was used instead of “reflux condenser”

Author Response

The authors have presented a study on preparation of a series of trimeric pyridinium salts assembled on a cyanuric acid core and their evaluation as antimicrobial agents. This structural motif has been already studied in several publications from the same group of authors. All synthetic procedures employed are quite standard and no significant improvement of bioactivity was achieved. Not only originality and novelty of this work is questionable but also the quality of the provided experimental data. My biggest concern is that no copies of NMR data were provided to support the purity and structure of the target compounds and moreover any data supporting their chemical formula such as HRMS or elemental analysis is missing. Unfortunately, I cannot recommend publication of this manuscript in its present form.

We thank the reviewer for the comments! It should be noted that all compounds obtained by us were synthesized for the first time, and the novelty of the method lies in its optimization for oxygen-containing platforms. Moreover, we observed a significant improvement in antibacterial activity compared to the previously synthesized series both on planktonic cells (by 2-16 times) and on biofilms (by 8 times), which can be seen from the MIC and MBC values in Table 2. Antifungal activity was also noted for new tris-QACs, superior to the commercial antiseptic octenidine.

Additional comments and questions:

  1. Copies of 1H and 13NMR spectra of all novel compounds must be provided (in a separated ESI file) to support their purity and structure. 

1H and 13C NMR spectra were added in Supporting Information

  1. HRMS or elemental analysis data must be provided for all novel compounds to support their chemical formula

HRMS data were added to experimental part (page 10-13).

  1. Schemes in the main text should be provided with better resolution

All schemes in the article are made in the ChemDraw professional 15.0 and already have the maximum possible resolution. Increasing the size of schemes will be left to the editor's choice. However, we do not consider this to be appropriate, since excessive enlargement of schemes can lead to their spreading beyond the column format.

  1. Yields should be added to reaction schemes. It is quite difficult to follow the discussion of yields while they are reported only in the materials and methods section.

We have made the appropriate corrections (page 3).

  1. NMR data should be double-checked. For example, it is unclear how can compound with 48 carbons (7a), which is symmetrical trimer with two identical carbons in pyridinium moiety has 18 signals in 13 NMR spectrum report.

According to our theory, such an effect is observed as a result of the isomerization of the aminopyridine ring. Thus, in pyridine, all five carbon atoms are different in 13C NMR. Isomerization is also seen in 1H NMR, where 4 proton signals in aromatics are observed instead of 2. This is confirmed by HRMS, where the presence of one or two charges is recorded, instead of the expected three. Therefore, we can observe small differences in the carbon chains in the NMR spectra, due to which double signals appear in a strong field (examples of isomers are given in Supporting Information Figure S2).

Relevant information was added on pages 9,10.

  1. This part of general information is unclear: “Only discrete or characteristic signals for the 1H NMR are reported.” Partial analytical data should not be used for characterization of novel compounds.

All necessary data were added in Supporting Information.

  1. On p.14 a procedure for water solubility determination is reported, but not the results were not discussed in the main text

The solubility of the new compounds was discussed briefly in the last paragraph of Results and Discussion (p. 9). Additionally, table of water solubility values was added in Supporting Information (Table S1).

  1. LogP can determined experimentally quite easily, especially for gram-scale amounts of substances. Why did the authors use calculated values instead? 

This method of calculating lipophilicity is simpler and has been used by many scientists before for similar purposes (list of publications is represented below).

https://doi.org/10.1016/j.ejmech.2020.113100

https://doi.org/10.1002/cmdc.202100409

https://doi.org/10.1021/acs.molpharmaceut.8b00177

  1. Extensive editing of English language required. For example, term “reverse refrigerator” was used instead of “reflux condenser”

We have made the appropriate corrections (page 9-10).

We thank the Reviewer for the comments!

Reviewer 3 Report

The article is devoted to solving the most important task of medical and organic chemistry – the creation of new promising antibiotics. The presented research is a continuation of the authors' work in this field. In comparison with previous works, this article describes an improved synthesis of new biologically active water-soluble of tris-quaternary ammonium compounds. Water-soluble compounds can be used in water-based biocide compositions, which is significantly less harmful then alcohol ones. A number of synthesized compounds demonstrated high antibacterial and antibiofilm activity.

There are the following comments on the work:

1. Page 4, 2-th paragraph, second  last sentence. There is a contradiction. It is indicated that "by-products were not detected". At the same time, the yield of 3a-c products was 51-60%. Are 40-50% impurities or unreacted substrates? Most likely, the low yield of 3 is just associated with the occurrence of side processes. At the beginning of the reaction, Isocyanuric acid is present in a large excess in the reaction mass. Therefore, at m = 1, replacement products of all chlorine atoms in compound 2 should be formed.

2. In the Materials and Methods section, it is desirable to specify mass spectra data for the obtained substances.

3. The authors of the article need to present NMR spectra of synthesized compounds in COMPLEMENTARY MATERIALS

4. It is necessary to conduct a broad study of the antibacterial activity of the substance 8b. This compound may be more promising compared to most compounds from the 7 series.

Author Response

The article is devoted to solving the most important task of medical and organic chemistry – the creation of new promising antibiotics. The presented research is a continuation of the authors' work in this field. In comparison with previous works, this article describes an improved synthesis of new biologically active water-soluble of tris-quaternary ammonium compounds. Water-soluble compounds can be used in water-based biocide compositions, which is significantly less harmful then alcohol ones. A number of synthesized compounds demonstrated high antibacterial and antibiofilm activity.

There are the following comments on the work:

  1. Page 4, 2-th paragraph, second  last sentence. There is a contradiction. It is indicated that "by-products were not detected". At the same time, the yield of 3a-c products was 51-60%. Are 40-50% impurities or unreacted substrates? Most likely, the low yield of 3 is just associated with the occurrence of side processes. At the beginning of the reaction, Isocyanuric acid is present in a large excess in the reaction mass. Therefore, at m = 1, replacement products of all chlorine atoms in compound 2 should be formed.

We are grateful to the Reviewer for this helpful remark! Indeed, interaction of cyanuric acid salt and trichloride cyanurate may led to by-products. However, after quenching the reaction process and distilling off the solvent, the residue was subjected to column chromatography and we were unable to identify individual by-products.

  1. In the Materials and Methods section, it is desirable to specify mass spectra data for the obtained substances.

HRMS data were added to experimental part (page 10-13).

  1. The authors of the article need to present NMR spectra of synthesized compounds in COMPLEMENTARY MATERIALS

1H and 13C NMR spectra were added in Supporting Information

  1. It is necessary to conduct a broad study of the antibacterial activity of the substance 8b. This compound may be more promising compared to most compounds from the 7 series.

We are grateful for this recommendation! Despite the occasional success with compound 8b, the 7th series were significantly more active against gram-negative bacteria, so we chose it for detailed study. The effectiveness of compound 8b can be explained by a combination of physicochemical properties, which may be the subject of a separate study that is beyond the scope of this manuscript.

We thank the Reviewer for the comments!

Round 2

Reviewer 1 Report

The statistical analysis is still not precise.

Actual or real photos of the antibacterial are not presented even in the supplementary materials.

minor errors

Author Response

Comments

1) The statistical analysis is still not precise.

We thank the reviewer for the remark! Statistical analysis (standard deviation) was added to all tables (see Tables 1-3). MICs and MBCs evaluations were performed in triplicate.

2) Actual or real photos of the antibacterial are not presented even in the supplementary materials.

Photos of MIC and MBC evaluations is added in SI (see pages S33-S34)

We thank the reviewer for the comments!

Reviewer 2 Report

The authors have performed some improvements to the manuscript but the quality of the presented experimental data is still questionnable.

Concerning HRMS data:

1) compound 3a: provided formula C15H24Cl3N3O3 is incorrect (there are six oxygens in the structure). The right one is C15H24Cl3N3O6 (exact mass 447.0731 Da). Then how can the found value be consistent with the wrong calculated data (477.0723)?

2) compound 7a: this molecule is a trication with Mw 876.6059 Da. HRMS-ESI method provides m/z values, which is around 876/3=292 Da. How can be the peak with m/z = 874 Da (described as [M + H - 3Cl]+) observed in experimental spectrum? Besides formula  [M + H - 3Cl] corresponds to tetrcation [M + H - 3Cl]4+, not monocation [M + H - 3Cl]+.

(!) Copies of all HRMS spectra must be provided in the ESI to demonstrate all reported peaks, including their intensities.

NMR data:

1) concerning copies of 1H spectra: at least one original NMR spectrum for compound 7 should be provided as raw data (for example  as mnova file, or *.fid). For the real samples especially with such large amount of aliphatic  protons and such purity the integrals are highly unlikely to match each other that perfectly (with two decimals eqal 00 almost in every spectrum) as we see in ESI.  

2) No 13C NMR spectra copies provided for compounds 7e,8a, 8b, 9b

3) No 1H and 13C NMR spectra copies provided for compounds 8c, 9a, 9c

Author Response

Comments

The authors have performed some improvements to the manuscript but the quality of the presented experimental data is still questionnable.

Concerning HRMS data:

1) compound 3a: provided formula C15H24Cl3N3O3 is incorrect (there are six oxygens in the structure). The right one is C15H24Cl3N3O6 (exact mass 447.0731 Da). Then how can the found value be consistent with the wrong calculated data (477.0723)?

We thank the reviewer for the remark! Indeed, provided formula of 3a was incorrect. That was simple typing error and we made the appropriate corrections.

2) compound 7a: this molecule is a trication with Mw 876.6059 Da. HRMS-ESI method provides m/z values, which is around 876/3=292 Da. How can be the peak with m/z = 874 Da (described as [M + H - 3Cl]+) observed in experimental spectrum? Besides formula  [M + H - 3Cl] corresponds to tetrcation [M + H - 3Cl]4+, not monocation [M + H - 3Cl]+.

(!) Copies of all HRMS spectra must be provided in the ESI to demonstrate all reported peaks, including their intensities.

We thank the reviewer for the comment! We provided HRMS scans for obtained salts, including calculated m/z (See SI pages S4-S31).

NMR data:

1) concerning copies of 1H spectra: at least one original NMR spectrum for compound 7 should be provided as raw data (for example  as mnova file, or *.fid). For the real samples especially with such large amount of aliphatic  protons and such purity the integrals are highly unlikely to match each other that perfectly (with two decimals eqal 00 almost in every spectrum) as we see in ESI.  

We provided examples of fid files in .zip (see file Fids of compounds 7-9).

2) No 13C NMR spectra copies provided for compounds 7e,8a, 8b, 9b

We provided all spectra in SI (see pages S22-S28), except 7e 13C NMR. This data was lost due to technical issues. Unfortunately, we do not have the reagents necessary for the synthesis of the desired compound at the moment.

3) No 1H and 13C NMR spectra copies provided for compounds 8c, 9a, 9c

We provided all NMR spectra mentioned in the comment.

We thank the reviewer for the comments!

Round 3

Reviewer 1 Report

Good work, I suggest it accept for publication after minor revision

1. Unify the references' style and format.

2. I suggest inserting some potential figures from the sublimity like "1H, 13C NMR, and HRMS spectra of Novel pyridinium tris-QACs" into the original manuscript.

3. Abstract still needs some potential results, quantitative results.

Minor corrections

Author Response

Comments and Suggestions for Authors

Good work, I suggest it accept for publication after minor revision

  1. Unify the references' style and format.

Corrected accordingly.

  1. I suggest inserting some potential figures from the sublimity like "1H, 13C NMR, and HRMS spectra of Novel pyridinium tris-QACs" into the original manuscript.

We thank the reviewer for the suggestion! We believe that the main point of this article is the antibacterial, antifungal and antibiofilm potential of the newly obtained compounds. All spectra images are available in SI and are carefully described in the materials and methods section.  

  1. Abstract still needs some potential results, quantitative results.

Abstract is enriched with quantitative results (see page 1)

We thank the reviewer for the comments!

Reviewer 2 Report

1) Copy of HRMS spectrum of 3a is missing.

2) The authors did not explain how can a molecule with three positive charges (arising from 3 pyridinium moieties) after additional protonation  demostrate in HRMS a peak corresponding to monocation [M+H]+ instead of tetracation. I assume that the structure of compounds 7,8 and was assigned incorrectly and the alkylation took place at NH-position of amine side chain insted of heterocyclic pyridine nitrogen. This means that the product molecule is not charged and therefore the regular monocation [M+H]+ was obseved in HRMS spectrum.  (!) Given this contradiction additional experiments such as single crystal X-ray analysis or acquiring 2D NMR spectra (HSQC, HMBC, NOESY) must be performed to support either authors' or reviewer's suggested structure.

Author Response

Comments and Suggestions for Authors

1) Copy of HRMS spectrum of 3a is missing.

HRMS spectrum of 3a was added in SI (see page S3)

2) The authors did not explain how can a molecule with three positive charges (arising from 3 pyridinium moieties) after additional protonation  demosntrate in HRMS a peak corresponding to monocation [M+H]+ instead of tetracation. I assume that the structure of compounds 7,8 and was assigned incorrectly and the alkylation took place at NH-position of amine side chain instead of heterocyclic pyridine nitrogen. This means that the product molecule is not charged and therefore the regular monocation [M+H]+ was obrseved in HRMS spectrum.  (!) Given this contradiction additional experiments such as single crystal X-ray analysis or acquiring 2D NMR spectra (HSQC, HMBC, NOESY) must be performed to support either authors' or reviewer's suggested structure.

We thank the reviewer for the comment! We acquired additional 2D NMR spectra for the structure confirmation (see pages S33-S35 for full description). The results supports our theory about novel tris-QACs structure. We highlighted key correlations for HMBC and COSY spectra (see page S33). Regarding the monocation in HRMS, this may be due to tautomerization (see page S36). Thus, novel tris-QACs can have a different number of charges, which can be seen in the HRMS. For example, for structure 7g (see page S20), two and three charged molecule are found.

We thank the reviewer for the comments!